# Mechanisms of PARP-Inhibitor-Resistance in BRCA-Mutated Breast Cancer and New Therapeutic Approaches

**DOI:** 10.3390/cancers15143642

**Published:** 2023-07-16

**Authors:** Sayra Dilmac, Bulent Ozpolat

**Affiliations:** 1Department of Nanomedicine, Houston Methodist Research Institute, Houston, TX 77030, USA; sdilmac@houstonmethodist.org; 2Houston Methodist Neal Cancer Center, Houston, TX 77030, USA

**Keywords:** BRCA1, BRCA2, PARP, PARPi, PARPi-resistance, breast cancer, combination therapies

## Abstract

**Simple Summary:**

Poly (ADP-ribose) polymerase (PARP) inhibitors treat breast and ovarian cancers. PARP-inhibition (PARPi), which affects cell survival by re-regulating DNA repair mechanisms, is known as one of the promising methods in terms of treatment protocols. However, resistance to PARP inhibitors causes disruptions to this treatment method. This study discusses the mechanisms that cause PARP-inhibitor-resistance and the possible therapeutic strategies to overcome them.

**Abstract:**

The recent success of Poly (ADP-ribose) polymerase (PARP) inhibitors has led to the approval of four different PARP inhibitors for the treatment of BRCA1/2-mutant breast and ovarian cancers. About 40–50% of BRCA1/2-mutated patients do not respond to PARP inhibitors due to a preexisting innate or intrinsic resistance; the majority of patients who initially respond to the therapy inevitably develop acquired resistance. However, subsets of patients experience a long-term response (>2 years) to treatment with PARP inhibitors. Poly (ADP-ribose) polymerase 1 (PARP1) is an enzyme that plays an important role in the recognition and repair of DNA damage. PARP inhibitors induce “synthetic lethality” in patients with tumors with a homologous-recombination-deficiency (HRD). Several molecular mechanisms have been identified as causing PARP-inhibitor-resistance. In this review, we focus on the molecular mechanisms underlying the PARP-inhibitor-resistance in BRCA-mutated breast cancer and summarize potential therapeutic strategies to overcome the resistance mechanisms.

## 1. Introduction

Breast Cancer 1 protein (BRCA1) is a tumor suppressor that is associated with repairing damaged DNA in double-stranded DNA breaks (DSBs) [1]. BRCA1 is encoded by a gene consisting of 22 coding exons distributed over 100 kb of genomic DNA. The BRCA1 gene encodes the 1863 amino acid sequence and more than 200 different germline mutations of this gene are related to cancer progression [2]. The question of why a gene such as BRCA1, which is required for DNA damage repair and proper DNA replication in all cells, increases the risk of breast and ovarian cancers has been opened to debate. Loss-of-function mutations are seen in many alleles that make the BRCA gene more susceptible to breast cancer. While realizing that only 45% of familial breast cancer cases were associated with BRCA1, the search for another breast-cancer-susceptibility gene emerged. Then, in 1995, the BRCA2 gene was identified on chromosome 13q12.3 [3,4].

BRCA2 mediates the recruitment of RAD51 filaments to the DNA double-stranded break site. The main task of BRCA2 is related to the HR mechanism and enables the assembly of the RAD51-BRCA2-DSS1 complex to generate undamaged HR [5]. In addition to its function in the DNA repair mechanism, BRCA2 inhibits tumor development by acting as a tumor suppressor [5,6]. 

Double-stranded DNA breaks may occur in DNA due to metabolic stress during DNA replication. With the occurrence of DNA damage, a DNA damage response (DDR) is activated with the help of ATM (ataxia-telangiectasia-mutated), ATR (Ataxia Telangiectasia-Mutated and Rad3-Related), and checkpoint (i.e., CHK1/2) kinases, leading to cell cycle arrest and DNA repair being initiated in cells [7]. DSB repair in the cell cycle is then successfully achieved, with the exception of the S and G2 phases. The repair of damage during the S and G2 phases also depends on the proximity of the damage to the replication fork. HR is one of the main avenues for damage repair in Late S and G2 [8].

## 2. Mutations in BRCA 1/2 Genes

BRCA1 and BRCA2 are important genes involved in homologous recombination (HR) repair, particularly in the repair of DNA DSBs, and are the causes of hereditary breast cancer. Mutations in the BRCA1 gene are closely associated with cancer formation [9]. Mutations in BRCA1 and BRCA2 have not only been associated with an increased risk of breast cancer [2,10] as they were later also found to increase susceptibility to ovarian, pancreatic, and prostate cancers. Although mutations of the BRCA1 and BRCA2 genes can cause hereditary cancers, they are extremely rare in sporadic breast cancers [11,12]. 

BRCA1 is known to localize to the replication fork, where it becomes hyperphosphorylated in response to DNA damage [2]. When DNA damage is encountered, BRCA1 is phosphorylated by ataxia-telangiectasia-mutated (ATM) kinase or ATM-related (ATR) kinase. BRCA phosphorylation is mediated by ATM kinase at Ser1387 residue, following ionizing radiation, and by ATR kinase at Ser1457, in response to ultraviolet radiation [13,14]. However, CHK2 (Checkpoint kinase 2), also known as the G2/M control kinase, phosphorylates BRCA1 at Ser988 when exposed to ionizing radiation [15]. Other BRCA1 sites that are phosphorylated in response to DNA damage are Ser1423 and Ser1524 [13,16]. Overall, data indicate that BRCA1 is phosphorylated at different phosphorylation sites by different kinases following DNA damage.

Studies have shown that BRCA1 and BRCA2 help repair the DSBs, initiate HR, and maintain genomic integrity, providing tumor-suppressing effects. BRCA1 and BRCA2 form complexes with the RAD51 (DNA repair gene) protein [17,18]. BRCA1 and BRCA2 play different roles in the repair of DSBs during HR. BRCA2 is thought to play a direct role and the loss of BRCA2 sensitizes cells to ionizing radiation due to defects in DSB repair. In addition to sensitivity, the propensity of the cell cycle checkpoint triggers apoptosis after DNA damage does not change [19,20].

In addition, BRCA2-deficient cells also show chromosomal breaks and abnormal mitotic activities. RAD51-deficient cells have similar phenotypes to BRCA2-deficient cells. This provides evidence that BRCA2’s interactions with RAD51 are essential for cell division and the maintenance of the chromosome structure. Studies showed that BRCA2 regulates the intracellular localization and function of RAD51 [21].

Traditionally, unlike other DSB repair pathways, the probability of error in HR repair is very low. BRCA1 promotes the end-resection of DSBs and initiates HR signaling with BRCA2 and PALB2 to stimulate the recruitment of RAD51 into the resected single DNA strand [22,23]. HR uses the newly replicated sister chromatid as a template to ensure the correct repair of the DNA helix [24].

## 3. Poly (ADP-Ribose) Polymerase 1 (PARP1) Is Critical for the Survival of BRCA-Mutated Cells

The repair of DNA damage is not only related to BRCA1 and BRCA2. BRCA1/2-mutant tumors also have HR-independent mechanisms for DNA repair through the activity of PARP1. PARP1 plays a very important role in the proper repair of DNA damage. In 2005, two studies showed that the inhibition of PARP1 activity is specifically cytotoxic in cells lacking functional forms of the tumor suppressors BRCA1 or BRCA2 [25,26]. 

PARP1 is a nuclear enzyme responsible for the regulation of many cellular processes through PARylation, including DNA damage, transcription, chromatin remodeling, the stabilization of replication forks, and the detection of unbound Okazaki fragments. During DNA damage, PARP1 is rapidly recognizing single-stranded breaks (SSBs) and DSBs. After binding to single-stranded DNA (ssDNA), PARP1 PARylates itself and other proteins, initiating the DNA repair signal and enabling other factors involved in this signal transmission to come into play at the right moment in time [23,27]. 

PARP1-inhibition can lead to DNA damage that, in the absence of BRCA1/2 function, triggers critical levels of genomic instability, mitotic catastrophe, and cell death [28]. Based on these findings, PARP-inhibition (PARPi) emerged as an important therapeutic option. The occurrence of HR mutations in high-grade serous ovarian cancers (HGSOCs), breast cancers (especially triple-negative breast cancers—TNBC), metastatic prostate cancers, and pancreatic cancers makes these tumors ideal candidates for PARPi therapy [29,30,31,32]. BRCA1 and RAD51C promoter methylation lead to HRD in HGSOCs. BRCA1 and RAD51C methylation also cause sensitization to PARP inhibitors and platinum-based chemotherapies [33,34]. In addition to the genomic aberrations seen in HRD tumors, the deletion of large genomic segments and a genome-wide loss of heterozygosity (LOH) can also be seen [35]. 

In HGSOC patients with a high LOH, a susceptibility to platinum and PARP inhibitors was detected independently of the BRCA mutations [36]. However, genomic damage in HRD cancer cells was present if homologous recombination repair (HRR) occurred, as well as a resistance to PARP inhibitors, making them suitable cell models to study PARPi-sensitivity [37,38]. 

## 4. PARPi Induces Synthetic Lethality in BRCA-Mutated Cells

Synthetic lethality occurs between two genes when there is no effect due to the disruption of either gene alone; but, when the disruption of both genes occurrs simultaneously, this results in loss of viability. The advantage of using synthetic lethality in cancer treatment is provided by correctly identifying synthetic lethal genes and characterizing ethical interactions. In addition to the mutation in the BRCA1 genes, the mutation in PARP1, the most critical component of the DNA repair mechanism, is one of the most successful examples of synthetic lethality [39]. 

Currently, there are four FDA-approved PARP inhibitors: (olaparib AZD2281 IC50 = 5 nM [40]; rucaparib AG15699 IC50 = 7.1 nM [41]; niraparib MK4827 IC50 = 3.8 nM [42]; and talazoparib BMN 673 IC50 = 0.57 nM [43]). They can be used for the treatment of patients [44,45,46]; and, there are an additional two inhibitors: (veliparib ABT888 IC50 = 3.3 nM [47] and pamiparib BGB290 IC50 = 1.3 nM [48]). They are undergoing clinical trials and evaluation [49]. 

PARP inhibitors prevent the binding of single-strand binding proteins (SSBs) that will repair DNA damage, thereby preventing damage to the replication fork [50]. In addition, DNA damage caused by PARP inhibitors is recognized by PARP1; then, PARP1 located in this area cross-links the DNA, triggering the collapse of replication forks. Thus, DSBs accumulate during the S phase of the cell cycle. These defects are not repaired by the HR system in HR-deficient tumor cells, triggering apoptosis (Figure 1) [51]. 

In most cell lines, only PARP-inhibition is incapable of inducing cell death [41]. PARP1 KO mice are known to be viable and fertile [42]. However, studies conducted in 2005 showed that breast cancer cells with mutations in the BRCA1 or BRCA2 genes are more susceptible to PARP-inhibition [21,22]. The BRCA1 and BRCA2 proteins are important in homologous recombination during the repair of DSBs. Mutations in the BRCA1 and BRCA2 genes impair downstream signaling molecules to efficiently repair DNA damage; as a result, apoptosis is induced in damaged cells [43]. On the other hand, BRCA1- and BRCA2-mutant cells are more dependent on PARP1 signaling for maintaining genomic integrity, making them more susceptible to the cytotoxic effects following PARP-inhibition [44].

In a previous study, we demonstrated that AZD2281 (olaparib) inhibits cell viability and induces autophagy/mitophagy in BRCA1- and BRCA2-mutant breast cancer cell lines. Furthermore, when BRCA1 and BRCA2 were down-regulated using shRNA in wild-type BRCA breast cancer cell lines, PARP inhibitors were highly effective in promoting the apoptosis of MDA-MB-231 and BT-20 triple-negative breast cancer cell lines, proving the concept that dual PARP1- and BRCA-inhibition is lethal for cells [45]. 

The most successful aspect of PARP-inhibition is limiting tumor progression due to a lack of proper homologous recombination for DNA repair, causing tumor cell death while normal cells remain unaffected [46,47]. A second treatment strategy involving the use of PARP inhibitors is the use of combination regimens, including cytotoxic chemotherapy, anti-angiogenic drugs, radiotherapy, immune therapy, and DNA damage-protein-inhibition [43].

PARP inhibitors also trap PARP enzymes in DNA. The resulting DNA–PARP complexes insert into the replication fork and block replication. Only HRR can reverse this situation [48]. BRCA1, BRCA2, and PALB2 (Partner and Localizer of BRCA2) are needed for this repair. BRCA1, BRCA2, and PALB2 are involved in the HRR and stabilize the PARPi-stopped replication fork. A loss of BRCA1/2 or PALB2 destabilizes the replication forks; this effect binds with PARP inhibitors during the synthetic lethality process, resulting in cytotoxic effects. This is exactly why the presence of PARP inhibitors and BRCA mutations are necessary, simultaneously, for an effective treatment via DNA damage-induced cell death [49]. 

In addition to the success of PARPi in the treatment of cancers with BRCA mutations, the increase in the frequency of use of these drugs in the clinic has revealed the problem of a resistance to PARPi, which became a new therapeutic challenge in cancer treatments. Therefore, understanding the mechanisms of resistance to PARPi is the first step in countering and overcoming resistance. In the following sections, we discuss the mechanisms of resistance to PARPi.

## 5. Resistance to PARPi

Inhibitors of PARP have been designed so that that they can be useful in the clinic by using their synthetic lethality feature. PARP inhibitors are known as the first synthetic-lethality-inducing agents allowed to be used in the clinic. Although PARP1 inhibitors are primarily used in the clinic to treat cancers with an impaired homologous recombination signaling pathway (especially BRCA1 mutation), they have recently been used as part of an adjunctive therapy in other solid cancers. This was based on the possibility that carriers without a HRD are susceptible to PARPi due to their BRCAness dependency. Possible mechanisms that may lead to the development of a resistance to PARPi are discussed in detail in the following sections. The most common PARPi resistance mechanisms detected in patient tumors include (a) dysregulated molecular signaling, (b) reverse mutations, (c) restoration of replication fork stability, and (d) increased drug efflux (Figure 2).

## 6. Molecular Mechanisms for Resistance to PARPi

Various mechanisms of resistance were identified and are being identified. Understanding these mechanisms is expected to lead to the development of novel strategies to overcome this acquired resistance to PARPi and sensitize BRCA-mutated breast cancer to PARPi in intrinsic resistant cases.

### 6.1. Dysregulated Molecular Signaling

Phosphatase and tensin homolog (PTEN) is a protein–lipid phosphatase that functions as a tumor suppressor and antagonizes the action of PI3K (Phosphoinositide 3-kinases), resulting in the inactivation of AKT (Protein Kinase B); this consequently inhibits cell growth and cell proliferation [52]. However, AKT activation can also induce BRCA1 expression, which then upregulates AKT’s downstream signaling pathways through a different signaling process [53]. Therefore, irregularities in BRCA levels may affect the PI3K/AKT pathway activation via PP2A (Protein Phosphatase 2). Damage to DSB repair can increase the proliferation of BRCA1-mutant cells via PI3K/AKT activation due to the loss of PTEN [54]. It is known that PTEN contributes to the regulation of RAD51. In fact, low levels of RAD51 have been shown in PTEN knockout mice and PTEN-/- human tumor cells. In addition, a decreased synthesis of nuclear RAD51 have also been shown in PTEN-mutant tumor cells [55]. Interestingly, endometrial cancer cell lines that have lost PTEN have problems with HR DNA repair; therefore, these cells are more sensitive to PARPi [56]. Based on these findings, a loss of PTEN, which is highly associated with BRCA1-mutated breast cancer, appears to promote PARPi-sensitivity. However, it should be noted that the presence of wild-type PTEN also confers a resistance to PARPi in breast cancer [57]. 

The most frequently activated cellular process in the repair of DNA damage is PARylation, catalyzed by PARP proteins. PARylation is catalyzed by both PARP1 and PARP2. Murai et al. have shown that the siRNA-mediated deletion of PARP1, but not PARP2, abolished olaparib cytotoxicity in prostate cells. Researchers have subsequently shown that PARP1 is the main factor responsible for PARPi-induced cytotoxicity [58]. PARP1 or PARP2 must be attached to DNA for effective PARylation to occur. Preventing PARP from adhering to DNA increases PARPi-resistance, even in the absence of HR. It is clear that the presence of PARP inhibitors, despite the presence of BRCA mutations, will not induce synthetic lethality if PARylation does not occur or if the PARP is prevented from adhering to DNA due to mutations [59].

### 6.2. Reverse Mutations

The BRCA mutations required for susceptibility to PARPi are reversible and restore protein function. The restauration of these mutations leads to PARPi-resistance. As a result of the restoration of BRCA1/2 function, the frameshift caused by mutation is lost and the open reading frame (ORF) is restored. This causes a full-length wild-type protein to be synthesized in its normal state, resulting in a genetic reversal of the mutation. Lin K.K et al. showed that in BRCA1- and BRCA2-mutated ovarian cancer cell lines, after treatment with cisplatin or PARP inhibitors, a protein in the mutated allele was restored, leading to platinum- and PARPi-resistance in these cells [60]. According to new studies, the return of the mutation found on BRCA1 and BRCA2 may cause resistance to PARPi and limit the success of the PARPi treatment. 

Studies demonstrated that a HR-deficiency leads to the activation of the non-homologous end joining (NHEJ) pathway; HR mutations, together with PARP inhibitors, lead to genomic instability and cell death. The decision on whether to use NHEJ or HR to repair DSBs is determined by multiple mechanisms, including cyclin-dependent kinase (CDK) activity and the activation of HR. NHEJ is active during the interphase; meanwhile, HR is active during the S and G2 phases of the cell cycle. In addition, during the S/G2 phases, HR is activated by the binding of the MRE11–RAD50–NBS1 (MRN) complex to the DSB terminals. To ensure synthetic lethality by PARPi in HR-deficient cells, NHEJs need to function properly. When NHEJ is blocked, the inhibition or downregulation of PARP in BRCA1, BRCA2, or ATM-deficient cell lines can escape cell death [61]. Therefore, a partially functional NHEJ mechanism in cancer cells can lead to a loss of PARPi-sensitivity.

PARylation is reversed by PAR glycohydrolase (PARG) via the breakage of PAR chains. Therefore, PARG works in the same framework as PARP inhibitors by preventing the accumulation of PAR chains. A loss of PARG in cell lines with BRCA1 and BRCA2 mutations appears to be a cause of PARPi-resistance. A loss of PARG results in a reduced DNA retention of PARP1 in PARP-inhibitor-treated cells and a partial amelioration of PARP1-induced DNA damage. Overall, the results of these experiments suggest that endogenous PARG activity is required for the cytotoxic effects of PARPi [62]. 

### 6.3. Restoration of Replication Fork Stability

BRCA1 and BRCA2 are also responsible for inhibiting the progression of the replication fork after DNA damage [63]. In the absence of BRCA1/2, MRE11 and MUS81 nucleases target the replication fork and cause its collapse and chromosomal aberrations. PTIP and EZH2, following the insertion of MRE11 and MUS81 into the replication fork, are known to cause PARPi-sensitivity [64]. Another factor responsible for the stabilization of the replication fork is RADX. While RADX prevents a MUS81-mediated fork collapse, it also prevents excessive remodeling of the replication fork, as mediated by RAD51 [65]. SMARCAL1, ZRANB3, and HLTF are involved in the modulation of the replication fork. They are required for the degradation of MRE11-dependent nascent DNA in cell lines with BRCA1 and BRCA2 mutations, as PARPi-resistance develops in the absence of SMARCAL1, ZRANB3, and HLTF [66].

Another factor impacting the replication stress is SLFN11; although, recent studies have shown that it does not directly affect the stability of the replication fork; it could rather arrest the replication fork. Replication fork arrest via SLFN11 also causes prolonged replication arrest in the S phase. This prolonged replication fork arrest induces replisome resolution and fork breakage, leading to hypersensitivity to PARPi [67].

### 6.4. Effect of the Increased Drug Efflux 

In addition to the resistance mechanisms specific to the DNA damage response, there are pharmacological factors regulating the response to PARPi. Recent studies suggest that the PARPi response is shaped by ATP binding cassette (ABC) transporters. The increased expression of Multidrug-Resistance Protein 1 (MDR1) [P-glycoprotein (PgP) efflux pump] in tumor cells, which is an important ABC transporter, increases the extracellular excretion of the drugs, resulting in a decrease in the effectiveness of PARP inhibitors [68]. In a BRCA1-mutant mouse model, PARPi-resistance developed when the Abcb1a and Abcb1b genes encoding the PgP pumps were upregulated [69]. Thus, that PARPi-resistance may be associated with an increased expression of drug efflux transporter genes; this resistance is mediated specifically by the Abcb1a/b genes. Rottenberg S. et al. showed that Abcb1a/b expression increased by two-fold up to eighty-five-fold in olaparib-resistant BRCA-mutated breast cancer cells [69]. Similarly, Abc1a/b expression in ovarian cancer cells has been shown to be associated with a resistance to olaparib and rucaparib. The administration of the MDR1 inhibitor tariquidar in ovarian and breast tumors has also been shown to sensitize tumors to PARPi [70,71]. In the meantime, a treatment with one of the Abcb1a/b inhibitors, verapamil or elacridar, was also able to reverse PARP-resistance [71]. However, these approaches have not been shown to reverse PARP-resistance in clinical trials. 

## 7. Approaches to Enhance the Effects of the PARPi Treatment

Resistance mechanisms against PARPi mediated through various intracellular mechanisms have been partially defined. Recent evidence suggests that other unknown factors could make tumor cells more sensitive to PARPi without triggering PARPi-resistance and supporting the use of PARPi in patients in a combinatorial therapeutic approach, along with chemotherapy and immunotherapies (Figure 3).

### 7.1. PARPi and Chemotherapy

The DNA repair mechanism is edited and regulated by genes involved in DNA damage. The idea of combining PARPi with chemotherapeutics inhibiting DNA repair was based on this fact. phase-I clinical trials have been initiated for the combinatorial use of DNA repair inhibitors, such as cisplatin, carboplatin, paclitaxel, and gemcitabine, with PARP inhibitors [72]. In a phase-I clinical trial, the combined use of 90 mg of paclitaxel and 200 mg of olaparib per week in metastatic triple-negative breast cancer showed significant myelosuppression [73]. In another study, in patients with high-grade breast cancer with a germline BRCA1/2 mutation, veliparib–carboplatin induced a superior response compared with that induced by a paclitaxel-placebo, carboplatin, and paclitaxel. While veliparib–carboplatin treated patients’ PFS (progression-free survival) time was 14.5 months, it was 12.6 months in the placebo/carboplatin group [74]. The combination of veliparib and paclitaxel showed promising results in PFS compared to chemotherapy alone. 

Xu et al. tested a veliparib and temozolomide (TMZ) combination in a phase-II study in patients with metastatic breast cancer and BRCA1/2-mutation-positive breast cancer. In the 28-day cycle, veliparib was used on the 1st and 7th days; initially, 40 mg was used but, then, it was reduced to 30 mg due to thrombocytopenia developing. Temozolomide was used in a 150 mg portion and, then, in a 200 mg portion, on the 1st and 5th days of the 28-day treatment cycle. The median PFS time was 3.3 months in patients with BRCA1/2 carriers and 1.8 months in non-carriers. In addition, the median PFS time was 2.7 months in platinum-naïve patients, compared to 1.9 months in platinum-treated patients. Platinum-naïve patients who were carriers of BRCA1/2 mutations were also shown to have a significantly prolonged PFS time compared to other groups. Overall, these studies suggest that the inhibition of PARP in the presence of BRCA-deficiency makes cancer cells more vulnerable to TMZ [75]. 

### 7.2. PARPi and Immune Checkpoint Inhibitors

Immunotherapy has been proven to be a promising treatment approach regarding breast cancer treatment in recent years. Commonly used immunotherapies include antibodies against programmed cell death ligand-1 (PD-L1) and cytotoxic T-lymphocytic-associated protein 4 (CTLA-4). It was also shown that the use of PARPi activates interferon I via cGAS-STING [76]. The activation of interferon I is also known to upregulate PD-L1 regarding anti-tumor immunity in breast cancer cells [77]. Furthermore, the PARPi treatment in TNBC cells increases the level of PD-L1 by causing immune activation [78]. 

Olaparib and talazoparib are known to increase PDL-1 protein expression in MDA-MB-231 and BT-549 cells. In this study, PD-L1 upregulation via PARPi treatment was found to reduce the therapeutic efficacy of PARP inhibitors via tumor-associated immunosuppression. Based on this study, the simultaneous inhibition of PARP and PD-L1 in the treatment of breast cancer tumors was proposed. In addition, clinical trials involving combinations of PARP inhibitors (olaparib, niraparib, and BGB-290) and PD-L1 or PD-1 antibodies were started [79]. 

Results from phase-II clinical trials demonstrated that the combination of PARP inhibitors (olaparib or niraparib) and anti-PD-L1 (durvalumab or pembrolizumab) antibodies demonstrated a good response against BRCA1/2-mutated breast cancer [80]. The results of many clinical studies on the effect of the combined use of PARP inhibitors with CTLA-4 antibodies and PD-L1 antibodies are being reviewed impatiently. 

### 7.3. PARPi and ATR/Chk1/Wee1-Inhibitor Therapy

ATR/Chk1/Wee1 inhibitors affect both HR and replication fork stability. Because of these effects, they were expected to reverse the resistance to PARPi and promote the re-sensitization of tumors. In BRCA1-deficient cells, ATR inhibitors restored HR-deficiency and reversed RAD51-dependent ‘stop replication’ fork protection [81]. It was recently shown that the combination of PARP inhibitors and ATR inhibitors creates a synergistic effect and provides an anti-tumor effect in PARPi-resistant BRCA1-mutant breast cancer models [82].

Inhibitors of Chk1, a downstream effector protein activated by ATR, have been shown to provide anti-tumorigenic effects when co-administered with PARP inhibitors in both BRCA-mutant and wild-type ovarian cancer models. Similar results were obtained when using a combination of Prexasertib (phase-I clinical trial), a Chk1 inhibitor, and olaparib in patients with PARPi-resistant, BRCA1-mutant HGSOC [83]. These promising research and clinical trials on ovarian cancer strengthen the possibility that Chk1 inhibitors may have a similar effect on breast cancer. The inhibition of Wee1, acting downstream of Chk1, has been shown to have a synergistic effect with PARP inhibitors in non-small cell lung cancer (NSCLC), ovarian cancer, and pancreatic cancer [84,85,86]. Like Chk1 inhibitors, Wee1 inhibitors are thought to be a promising part of tumor-targeted therapy in breast cancer utilized by potentiating the effect of PARPi.

### 7.4. PARPi and BRD4/BET Inhibitor Therapy

Bromodomains (BRDs) are proteins responsible for histone acetylation, mediating the binding of transcriptional regulators to acetylated histones. The bromodomain and extra-terminal domain (BET) family, one of eight subfamilies of the BRDs, contain proteins responsible for the transcriptional activation (recruitment of positive transcription elongation factor (P-TEFb)) and control of RNA polymerase II [87]. Bromodomain 4 (BRD4) is also a member of the BET protein family that is responsible for epigenetic gene regulation. It has been observed that cells resistant to PARP inhibitors in breast and ovarian cancers are re-sensitized by BRD4-inhibition (BRD4i) or BET-inhibition (BETi). BET-inhibition has been shown to suppress HR-related genes, including BRCA1, RAD51, and CtIP (C-terminal binding protein (CtBP) interacting protein), thereby causing an HR-deficiency. The combination of PARP inhibitors (olaparib) and BRD4 inhibitors (JQ1) has been shown to increase the anti-tumorigenic effect in ovarian and breast cancers in vitro and in vivo [88,89,90]. 

### 7.5. PARPi and CDK12 Inhibitor Therapy

Cyclin-dependent kinase 12 (CDK12) regulates transcription by acting on RNA polymerase II and is also involved in RNA splicing, translation, DNA damage response (DDR), cell cycle progression, and cell proliferation. CDK12-inhibition (CDK12i) reduces the expression of HR-related genes and impairs HR repair [91,92,93,94]. In high-grade serous ovarian cancer and breast cancer models, the combination of CDK12i and PARPi led to synthetic lethality by re-inducing HR. Based on these studies, it is thought that the combinatorial use of CDK12i and PARPi will be effective against the resistance of HR to PARPi [94,95]. 

## 8. Conclusions

The PARP family of proteins, discovered more than half a century ago, proved to have very important functions in terms of many cellular processes, including transcription, cell death, and DNA repair. Replication stress and mutations in DNA damage repair genes, which are frequently observed in cancer cells, and have led to new ideas for treatment strategies that are PARP-centered. In particular, the knowledge about the roles of PARP1 in DNA repair has led to the development of PARP inhibitors for the treatment of cancers with BRCA mutations. However, there is also strong evidence that the effect of PARPi is not limited to the inhibition of PARP1’s catalytic activity. With the increase in the clinical use of PARPi, especially in breast and ovarian cancers, resistance to PARP inhibitors, unfortunately, limits the clinical use. Encountering this challenge leads to investigations into identifying strategies to increase the effect of PARPi or to reverse the resistance against PARP inhibitors. Recent studies revealed that the sensitivity of certain tumors to PARPi can be increased; or, resistance can be overcome, especially by targeting pathways other than HR, including PARP1, both directly and indirectly. In addition, studies have revealed the different anti-tumorigenic effects of using PARP inhibitors in combination with other anti-cancer agents to achieve significant tumor regression. Ongoing clinical trials have revealed the need for combination therapies to escape PARPi-resistance. Among these promising combinations are PARPi and chemotherapy, PARPi and immune checkpoint inhibitors, PARPi and ATR/Chk1/Wee1 inhibitors, PARPi and BET inhibitory therapy, and PARPi and CDK1/2 inhibitor therapies. Although preliminary data suggest positive responses and clinical benefits, the short duration of the response is among the main clinical problems that make it difficult to use combined therapies. Since PARP inhibitors were approved in 2014, continuous efforts have been made to establish biomarkers for PARP-inhibitor-susceptibility. Currently, BRCA1/2 mutations remain the most common biomarkers. In addition, in 2019, the HRD test was approved as a biomarker for using niraparib in patients with advanced ovarian cancer. In addition, tumors with high KRAS expression that are sensitive to PARPi treatment and increased AKT1 expression after PARPi treatment may be associated with PARPi-resistance. Further investigation into molecular response markers is needed to determine which of the combined therapies with PARPi will be more effective. In this way, the most effective treatments that will provide the maximum benefit for cancer patients will emerge. For this reason, there is a need for more advanced studies to unveil the mechanisms of resistance to PARPi and to identify the best treatment combinations.

## Figures and Tables

**Figure 1 cancers-15-03642-f001:**
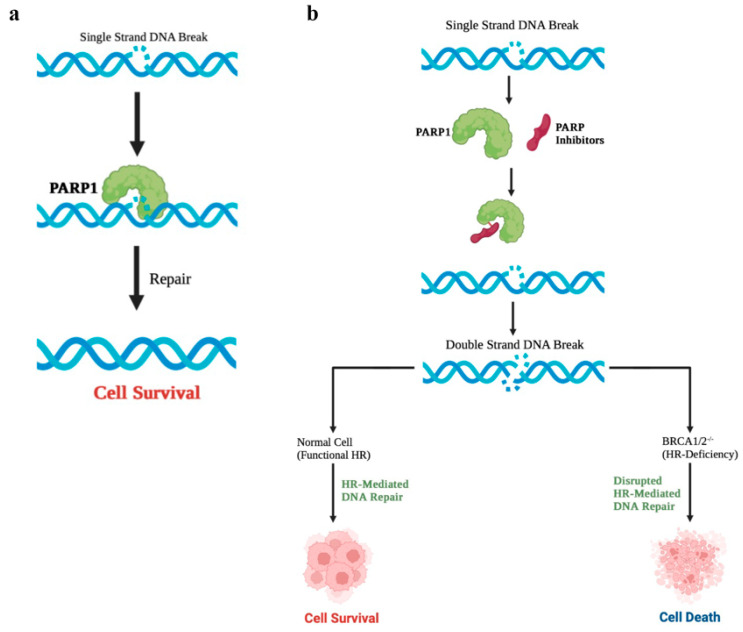
Mechanism of PARP inhibitors. (**a**) Endogenous single-strand breaks (SSBs) are frequently seen in rapidly proliferating cells, such as cancer cells. A SSB is repaired with the help of PARP. Effectively repairing SSBs inside the cell is important for cell survival. (**b**) PARP inhibitors inhibit the binding of PARP to DNA breaks and, thus, prevent the repair of RCC. Unrepaired SSBs can degenerate into double-strand breaks (DSBs) that are toxic to cells; homologous recombination (HR) is the primary pathway for repairing such DNA breaks during cell replication. HR-proficient cells can repair SSB-derived DSBs to ensure genome stability and cell survival; whereas, HR-deficient cells cannot repair DSBs and apoptosis occurs.

**Figure 2 cancers-15-03642-f002:**
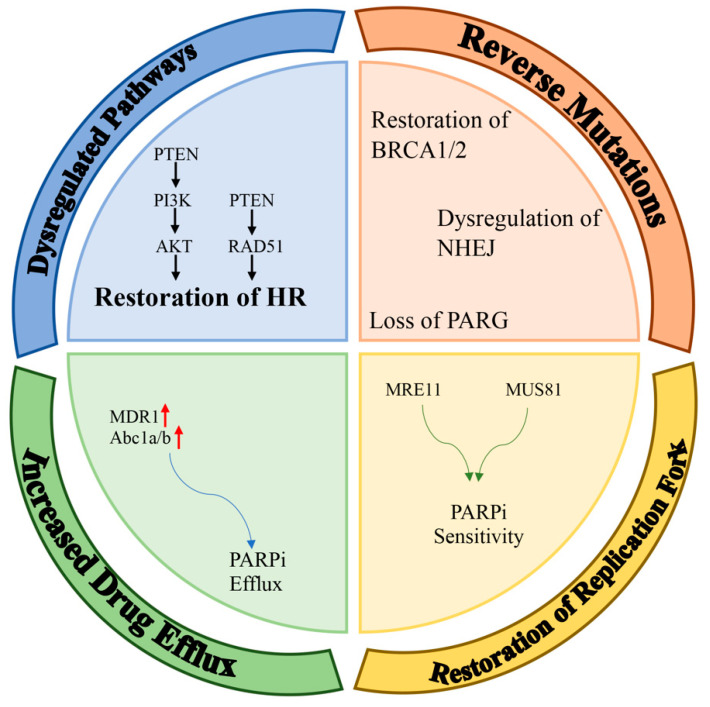
Mechanisms of resistance to PARPi. Potential mechanisms of PARPi-resistance can be classified into four main groups: dysregulation of pathways; restoration of HR; reverse mutations (e.g., restoration of BRCA1/2, dysregulation of NHEJ, loss of PARG); restoration of replication forks; and increased drug efflux (red arrows means up-regulation).

**Figure 3 cancers-15-03642-f003:**
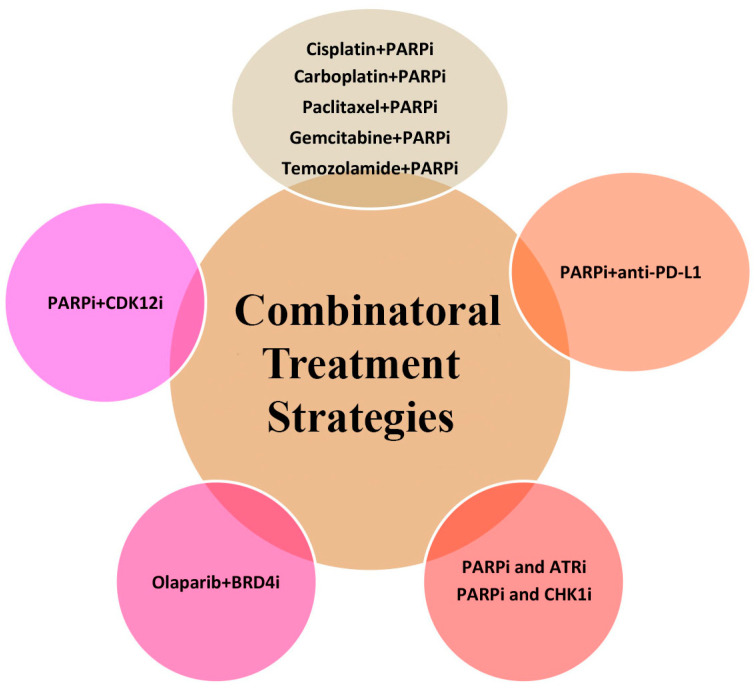
Potential novel combinatorial treatment strategies with PARPi. Combinations with chemotherapy; combinations with immunoregulators; and combinations with molecules associated with DNA repair, BET inhibitors, or immune checkpoint kinase inhibitors are among the promising strategies that may be used in combination with the PARPi treatment.

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
