# Peer review of "Mechanisms of PARP-Inhibitor-Resistance in BRCA-Mutated Breast Cancer and New Therapeutic Approaches"

_cancers, 2023, doi:10.3390/cancers15143642_

Round 1

Reviewer 1 Report

Review

cancers-2491535

“MECHANISMS of PARP INHIBITOR RESISTANCE IN BRCA-MUTATED BREAST CANCER and NEW THERAPEUTIC APPROACHES”

Sayra Dilmac and Bulent Ozpolat

Dear Editor, Dear Authors,

Recently, an idea of synthetic lethality has been extensively explored to increase the efficiency of cancer treatment. It has been found that PARP inhibitors (PARPi) are more potent when applied to homologous recombination-deficient tumours. However, frequent resistance to PARPi is developed, which mitigates the efficiency of the cancer therapy. In the following review, the authors provide a comprehensive summary of the mechanisms underlying the resistance to PARPi and provide therapeutic strategies to overcome these effects.

However, despite the increased interest in the topic, the following review requires some editing and clarifications, which I believe will improve significantly the content and increase the scientific value of the article. Please, find below my detailed suggestions and comments:

1.      It is recommended to extend the information about the BRCA2 gene and its product in the Introductory section.

2.      The chapter “Mutations in BRCA 1/2 genes” is unstructured and required some editing in order to clarify the importance of BRCA1 and BRCA2 mutations. However, in order to emphasize these, authors should also focus on introducing to the reader the molecular functions of BRCA1 in DNA end resection as well as the function of BRCA2 in homologous recombination (HR).

3.      Along with the above suggestions, it will be also appreciated if a small paragraph about DSB repair pathways and their efficiency throughout the cell cycle is included.

4.      It is suggested Chapter 3 to include a more comprehensive explanation of the PARP1 functions, essential for DSB repair in HRD background.

5.      There is no clear definition in the text of what exactly “synthetic lethality” is, and how this concept is exploited to potentiate cancer treatment.

6.      The text will read more smoothly if a brief explanation of the content of Figure 2 is provided before the detailed explanation.

7.      An illustration summarizing the content of the Chapter 7 will help for a better understanding of the provided molecular mechanisms.

Minor issues

8.      The complete name of the ATR is Ataxia Telangiectasia-Mutated and Rad3-Related.

9.      It is good to abbreviate IR after the first appearance.

10.  Scientific names of the included PARPi should be provided. Information about IC50 and the trapping activity for every inhibitor will be useful for the explanation of the mechanism of the inhibitor action.

11.  The suggested mechanisms, provided in Figure2 (Reverse Mutations), could be separated as distinct mechanisms.

There are no concerns about the quality of the English Language.

Author Response

Reviewer 1:

Dear Editor, Dear Authors,

Recently, an idea of synthetic lethality has been extensively explored to increase the efficiency of cancer treatment. It has been found that PARP inhibitors (PARPi) are more potent when applied to homologous recombination-deficient tumours. However, frequent resistance to PARPi is developed, which mitigates the efficiency of the cancer therapy. In the following review, the authors provide a comprehensive summary of the mechanisms underlying the resistance to PARPi and provide therapeutic strategies to overcome these effects.

However, despite the increased interest in the topic, the following review requires some editing and clarifications, which I believe will improve significantly the content and increase the scientific value of the article. Please, find below my detailed suggestions and comments:

  1. It is recommended to extend the information about the BRCA2 gene and its product in the Introductory section.

We appreciate your suggestion. We included information about BRCA2 gene and its product. All new revisions are highlighted in red in the text.

  1. The chapter “Mutations in BRCA 1/2 genes” is unstructured and required some editing in order to clarify the importance of BRCA1 and BRCA2 mutations. However, in order to emphasize these, authors should also focus on introducing to the reader the molecular functions of BRCA1 in DNA end resection as well as the function of BRCA2 in homologous recombination (HR).

We expanded the part as suggested. However, due to limited space we focused more on the resistance mechanism to PARP inhibitors and the mechanism to overcome resistance and PARP inhibition mechanisms' molecular basis.

  1. Along with the above suggestions, it will be also appreciated if a small paragraph about DSB repair pathways and their efficiency throughout the cell cycle is included.

We included information about the DSB Repair pathway.

  1. It is suggested Chapter 3 to include a more comprehensive explanation of the PARP1 functions, essential for DSB repair in HRD background.

We explained the function of PARP1 in BRCA mutant cells. Again, because our aim is to explain the resistance mechanisms to PARP inhibitors and the mechanism to overcome resistance. Thefore, due to space limitation we could not provide extensive information about in some areas.

  1. There is no clear definition in the text of what exactly “synthetic lethality” is, and how this concept is exploited to potentiate cancer treatment.

The part regarding the concept of synthetic lethality was expanded to provide better description about the topic.

  1. The text will read more smoothly if a brief explanation of the content of Figure 2 is provided before the detailed explanation.

Brief information was added before Figure 2.

  1. An illustration summarizing the content of the Chapter 7 will help for a better understanding of the provided molecular mechanisms.

We included a new figure to better describe the concept in Chapter 7.

Minor issues

  1. The complete name of the ATR is Ataxia Telangiectasia-Mutated and Rad3-Related.

The complete name of ATR was corrected.

  1. It is good to abbreviate IR after the first appearance.

IR abbreviation was included in the text.

  1. Scientific names of the included PARPi should be provided. Information about IC50 and the trapping activity for every inhibitor will be useful for the explanation of the mechanism of the inhibitor action.

We included the scientific names of the PARPi and their IC50 values.

  1. The suggested mechanisms, provided in Figure2 (Reverse Mutations), could be separated as distinct mechanisms.

We four mechanisms including reverse mutations in the figure.

Reviewer 2 Report

The manuscript provides an exploration of PARP inhibitors (PARPi) in cancer treatment, focusing on their use in targeting cancer cells with BRCA mutations. It acknowledges the challenge of resistance to PARPi and discusses potential mechanisms underlying resistance. Various approaches to overcome resistance and enhance treatment outcomes are also addressed. The manuscript emphasizes the need for further research to improve our understanding of PARPi resistance mechanisms and to identify optimal treatment combinations. Additionally, it highlights the significance of combination therapies in addressing PARPi resistance and improving treatment outcomes in cancer.

To enhance the manuscript, the following improvements can be made:

1) Provide a more in-depth exploration of the mechanisms of resistance to PARPi, including the underlying genetic and molecular factors. Analyze whether the current four FDA-approved PARPi and the two undergoing trials exhibit the same mechanism of action or if there are differences among these inhibitors. Clarify whether the discussed resistance mechanisms, such as dysregulated pathways, reverse mutations, increased drug efflux, and restoration of replication fork, are acquired after PARPi treatment or intrinsic to the cancer cells before PARPi treatment.

2) Besides the combinatorial therapeutic approaches, could the authors explore and discuss potential future directions in PARPi research to enhance treatment efficacy. This can include investigating novel combination therapies, emerging targets, and strategies to overcome resistance. Such discussions will provide readers with valuable insights into ongoing developments in the field.

3) Expand on the investigation of molecular response markers mentioned in the discussion section. Discuss potential biomarkers for predicting response to PARPi treatment, such as assessing DNA damage repair proficiency, genomic instability, and specific gene mutations beyond BRCA1/2. This will provide a more comprehensive understanding of predictive factors for PARPi efficacy.

By incorporating these improvements, the manuscript will offer a more detailed analysis of PARPi resistance mechanisms, provide insights into future research directions, and explore additional biomarkers for predicting treatment response.

Minor:

Line 205, Olaparip, should be Olaparib.

Author Response

Reviewer 2:

The manuscript provides an exploration of PARP inhibitors (PARPi) in cancer treatment, focusing on their use in targeting cancer cells with BRCA mutations. It acknowledges the challenge of resistance to PARPi and discusses potential mechanisms underlying resistance. Various approaches to overcome resistance and enhance treatment outcomes are also addressed. The manuscript emphasizes the need for further research to improve our understanding of PARPi resistance mechanisms and to identify optimal treatment combinations. Additionally, it highlights the significance of combination therapies in addressing PARPi resistance and improving treatment outcomes in cancer.

To enhance the manuscript, the following improvements can be made:

1) Provide a more in-depth exploration of the mechanisms of resistance to PARPi, including the underlying genetic and molecular factors. Analyze whether the current four FDA-approved PARPi and the two undergoing trials exhibit the same mechanism of action or if there are differences among these inhibitors. Clarify whether the discussed resistance mechanisms, such as dysregulated pathways, reverse mutations, increased drug efflux, and restoration of replication fork, are acquired after PARPi treatment or intrinsic to the cancer cells before PARPi treatment.

Although we aimed to focus more on the aquired resistance mechanisms developed during the treatment with PARPi drugs in fach this is related to exixtance of preexistings resistant cloes or selection/enrichment of resistant clones during the treatment, thus, in fact both intrinsic and aquired resistant mechanims are realted to each other other an aquired resistance mechanims provide information on the intrinsic drug resistance. In addition, due to word limitations imposed the the journal we were not able to expand some of the sections.

2) Besides the combinatorial therapeutic approaches, could the authors explore and discuss potential future directions in PARPi research to enhance treatment efficacy. This can include investigating novel combination therapies, emerging targets, and strategies to overcome resistance. Such discussions will provide readers with valuable insights into ongoing developments in the field.

In the part 7.0 (Approaches to Enhance the Effects of PARPi Treatment), we provided potential new treatment targets of PARPi are discussed, as you suggested. In this section, due to word limitations we only mentioned the mechanisms that help overcome PARPi resistance.

3) Expand on the investigation of molecular response markers mentioned in the discussion section. Discuss potential biomarkers for predicting response to PARPi treatment, such as assessing DNA damage repair proficiency, genomic instability, and specific gene mutations beyond BRCA1/2. This will provide a more comprehensive understanding of predictive factors for PARPi efficacy.

We added a section on possible potential biomarkers to evaluate the response to PARPi treatment.

By incorporating these improvements, the manuscript will offer a more detailed analysis of PARPi resistance mechanisms, provide insights into future research directions, and explore additional biomarkers for predicting treatment response.

Minor:

1) Line 205, Olaparip, should be Olaparib.

Thank you for your correction. We corrected the word.
